# The Treatment of Lung Involvement in Systemic Sclerosis

**DOI:** 10.3390/ph14020154

**Published:** 2021-02-13

**Authors:** Barbara Ruaro, Marco Confalonieri, Marco Matucci-Cerinic, Francesco Salton, Paola Confalonieri, Mario Santagiuliana, Gloria Maria Citton, Elisa Baratella, Cosimo Bruni

**Affiliations:** 1Department of Pulmonology, University Hospital of Cattinara, 34149 Trieste TS, Italy; marco.confalonieri@asugi.sanita.fvg.it (M.C.); francesco.salton@gmail.com (F.S.); paola.confalonieri.24@gmail.com (P.C.); mario@marionline.it (M.S.); gloriacitton@gmail.com (G.M.C.); 2Department of Experimental and Clinical Medicine, Division of Rheumatology, University of Firenze, 50121 Firenze FI, Italy; marco.matuccicerinic@unifi.it (M.M.-C.); cosimobruni85@gmail.com (C.B.); 3Department of Radiology, Cattinara Hospital, University of Trieste, 34127 Trieste TS, Italy; elisa.baratella@gmail.com

**Keywords:** systemic sclerosis, scleroderma, interstitial lung disease, pulmonary function tests, high-resolution computed tomography

## Abstract

Systemic sclerosis (SSc) patients are often affected by interstitial lung disease (ILD) and, although there have been recent treatment advances, it remains the leading cause of death among SSc, with a 10-year mortality up to 40%. African Americans and subjects with diffuse cutaneous SSc or anti-topoisomerase 1 antibodies are most commonly affected. Currently, early ILD diagnosis can be made, and it is pivotal to improve the prognosis. The diagnostic mainstay test for SSc-ILD is high-resolution computed tomography for the morphology and pulmonary function tests for the functional aspects. Treatment planning and intensity are guided by the disease severity and risk of progression. Traditionally, therapy has depended on combinations of immunosuppressants, particularly cyclophosphamide and mycophenolate mofetil, which can be supplemented by targeted biological and antifibrotic therapies. Benefits have been observed in trials on hematopoietic autologous stem cell transplantation for patients with progressive SSc, whilst lung transplantation is reserved for refractory SSc-ILD cases. Herein, recent advances in SSc-ILD treatment will be explored.

## 1. Introduction

Scleroderma or Systemic Sclerosis (SSc), a disease characterized by fibrosis, vasculopathy, and inflammation, may affect different organ and systems, with severe prognostic implications [1,2]. When SSc pathogenetic processes manifest at lung level [3], pulmonary disease may manifest both as interstitial lung disease (ILD) and/or pulmonary arterial hypertension (PAH) [4,5]. The European Scleroderma Trials and Research (EUSTAR) group reported that 53% of cases with diffuse cutaneous SSc (dcSSc) have ILD, as do 35% of cases with limited cutaneous SSc [6]. Moreover, high-resolution computed tomography (HRCT) evidences interstitial abnormalities in 90% of SSc patients [7], and pulmonary function tests (PFT) showed alterations in 40–75% [8]. There has been no significant change in SSc mortality rate over the past 40 years [9,10], although an increase in mortality due to ILD and PAH [11,12] is significant, a decrease in deaths due to renal crisis has been recorded [13]. Nowadays, ILD and PAH are the two leading causes of death in SSc, accounting for 33% and 28% of deaths, respectively [10,11,12]. The survival of systemic sclerosis-related interstitial lung disease (SSc-ILD) patients is reported to be 29–69% at 10 years [9,12]. Early autopsy studies demonstrated that up to 100% of patients had parenchymal involvement [14]. Considering the frequency and the prognosis of SSc-ILD patients, it is essential to attempt to identify pulmonary disease early, at a potentially reversible stage [15]. 

Unfortunately, there are limited treatment options for this manifestation, given that the paucity of high-quality, randomized, controlled trials specifically targeting SSc-ILD are scanty, and, historically, studies have favored cyclophosphamide (CYC) for the treatment of SSc-ILD, as also suggested in the most recent European League against Rheumatism (EULAR) recommendations [16,17,18,19]. The most recent and supportive data showed the positive effect of nintedanib, a multi-tyrosine kinase inhibitor, as a significant inhibitor of progressive functional decline [20]. Innovative proposals have also recently been made on the basis of clinical and preclinical evidence for rituximab (RTX), tocilizumab (TCZ), and pirfenidone (PIRF), as well as hematopoietic stem cell transplantation and lung transplantation [21]. However, the safety and efficacy of emerging experimental therapies for SSc-ILD require further investigation. The aim of this review is to summarize the state-of-the art in SSc-ILD treatments.

## 2. Management Principles

As SSc-ILD is a very heterogeneous disease, management tends to differ according to the profile of the patient. Furthermore, with the advent of the new aforementioned treatment options, it is pivotal to detect ILD [22,23,24] as early as possible and also to assign the right treatment as early as possible [7,19]. Toward this aim, precise and objective ILD classification tools that allow for patient stratification at ILD detection and diagnosis play a major role [25]. Indeed, patients must be classified by a severity assessment of ILD at diagnosis, performed by HRCT and PFT, and then by the evaluation of the risk of ILD progression [25,26]. 

The HRCT variables predictive of mortality and ILD progression in SSc–ILD were studied and reported in a recent meta-analysis of 27 studies, which concluded that the extent of disease on HRCT was an independent predictor of both mortality and ILD progression [27].

It is a must to detect the subset of clinical ILD patients with progressive disease, defined as a decline in Forced Vital Capacity (FVC) levels of >10% from baseline or a ≥ 5% to < 10% relative decline in FVC and a ≥ 15% relative decline in Diffusion Lung Capacity of Carbon monoxide (DLCO) over 12 months [28]. Despite this cut-off being proposed for clinical trials and applied also in clinical practice, smaller changes may also be of clinical importance, in particular worsening symptoms attributable to ILD [25,26,29]. DLCO alone was also one of the most consistent predictors of mortality, a finding that may well help in the identification of patients with a poor prognosis, even if these preliminary findings should be confirmed and expanded by further rigorous studies [25,26,27]. The likelihood of progression, comorbidities, and toxicity risks and current data on efficacy are often the basis for decisions taken to initiate or advance treatment [30]. The goal of treating clinical SSc-ILD is the stabilization or prevention of progressive disease.

## 3. Treatment Options 

The 2017 EULAR recommendations for the treatment of SSc state that the physician’s assessment of symptoms, disease severity, and/or disease progression form the basis for decisions to initiate SSc-ILD treatment, with a tailored risk–benefit evaluation especially in progressive SSc-ILD patients [18]. There was also a recommendation for the use of CYC and hematopoietic stem-cell transplantation for SSc-ILD patients. After the release of this recommendations, evidences for a positive effect of mycophenolate mofetil (MMF) and nintedanib have also become available [17,20]. Several studies have also been reported that tocilizumab and rituximab might be able to slow down ILD progression [21]. A summary of the treatment options further discussed is presented in Table 1.

## 4. Conventional and Biologic Immunosuppressants

SSc is a connective tissue disease where inflammation and immune abnormalities play a central role [64,65,66,67]. The immune system, especially B and T lymphocytes, is involved in fibroblast activation and fibrogenesis as they secrete proinflammatory and profibrotic cytokines and growth factors [64,65,66,67]. That is why traditional immunosuppressant, e.g., CYC, MMF and azathioprine (AZA) have been so far considered the milestones of SSc-ILD treatment. 

### 4.1. Cyclophosphamide

CYC is the most commonly used immunosuppressant, and it has been tested in numerous open-label studies, as well as in a few randomized control trials (RCT) [68]. CYC is recommended as first-line therapy in SSc–ILD patients in the EULAR guidelines [18].

In the Scleroderma Lung Study (SLS) I, 1-year course of oral CYC up to 2 mg/kg/day showed a statistically significant but small improvement in FVC (2.5% improvement) vs. placebo and little sustained benefit after discontinuation [16]. Similar results were not confirmed in the Fibrosing Alveolitis in Scleroderma Trial (FAST), which reported no statistically significant difference between the placebo and CYC group [31]. The clinical significance of this is modest, yet real improvement in FVC is still under debate and it seems that there will be decades of pros and cons. Noteworthy is the fact that the SLS I patients most likely had a stable SSc-ILD, as only 15% of them needed to restart an immunosuppressive treatment after the end of the study [16].

The SLS II (head-to-head comparison of oral CYC up to 2 mg/kg/day for 1 year plus 1 year of placebo versus MMF at up to 1.5 g twice daily for 2 years) showed that the benefits of MMF on FVC and on improvement in dyspnea were similar to those obtained with oral CYC at 2 years (MMF 2.2%, CYC 2.9%), with a safety profile favoring MMF [17].

In conclusion, it seems that CYC can either stabilize worsening SSc-ILD or modestly improve stable SSc-ILD; these data were also confirmed in a recent comparison between intravenous and oral CYC administration analyzing patients derived from the SLS1, SLS2, and EUSTAR cohorts. These results showed non-different effect on FVC change and ILD progression for the two routes of administration, despite a significantly lower CYC dosages in the intravenous group and a significantly different safety profile [32].

### 4.2. Mycophenolate Mofetil

MMF inhibits lymphocyte proliferation and is a safer, less toxic alternative to CYC for the treatment of SSc–ILD. Indeed, the safety and efficacy of MMF in SSc–ILD patients has been reported in several case series, uncontrolled studies, and, more recently, 2 meta-analyses [17,33,40,41]. Recently, the SLS II study, which reported on SSc–ILD patients treated with MMF for 2 years or CYC for 1 year followed by one year of placebo, showed that both treatment regimens led to a significant improvement in the pre-specified measurements of lung function over the 2-year study period. However, even if MMF was better tolerated and had lower toxicity levels, the hypothesis that it would be more efficacious at 24 months than CYC was not confirmed [17]. Although these data support the potential clinical efficacy of both CYC and MMF for progressive SSc–ILD, there is a possible preference for MMF due to its better tolerability and toxicity profile [17]. Lastly, Owen et al. demonstrated that MMF therapy was associated with a clinical stability for up to 36 months and lower frequency of early adverse events compared to AZA for SSc–ILD patients with a decline in pulmonary function [42].

### 4.3. Azathioprine

Although some small case series and retrospective studies suggested that AZA could be used as maintenance immunosuppressive treatment for SSc–ILD [34,35], a randomized unblinded clinical trial comparing the use of CYC and AZA (a purine analog) as first-line treatment did not evidence the efficacy of AZA in the treatment of SSc–ILD [36]. In addition, the very recent study by Owen et al. showed the better efficacy and tolerability of MMF versus AZA in the management of SSc–ILD [42].

### 4.4. Rituximab

A few case reports and open-label uncontrolled studies reported an improvement in SSc–ILD with RTX. Indeed, RTX therapy in SSc has gained favor after reports on its promising effects on both ILD and skin thickening [52,53,69]. The largest observational study available so far was published by the EUSTAR group and included 254 SSc patients treated with RTX, showing a good safety profile, steroid sparing agent potential and good efficacy profile on the skin but not on the lung. At pulmonary level, the combination of RTX + MMF determined a significant reduction in FVC decline over time, compared to monotherapy, therefore hypothesizing a higher promising potential for the combination treatment [54]. A similar safety profile and potential beneficial effect was also confirmed in an observational cohort receiving biosimilar RTX [55]. 

A recent open-label, randomized, controlled trial of head-to-head RTX vs. monthly pulse CYC reported on a population of 60 early, treatment naïve, anti-SCL-70+, dcSSc with ILD patients receiving either CYC or RTX. At the end of 6 months, the authors observed that FVC improved from 61.3% to 67.5% in the RTX group, whilst it did not in the CYC group (59.3% to 58.1%) [37]. The currently ongoing Rituximab versus Cyclophosphamide in Connective Tissue Disease-ILD (RECITAL) study (NCT01862926) is investigating the same topic in a larger cohort of connective tissue diseases related ILD, with a longer follow-up (48 weeks) [70].

### 4.5. Tocilizumab

The first two studies on TCZ, an anti-IL-6 soluble receptor monoclonal antibody, reported inconclusive results [48,49]. TCZ was administered in a phase 2 study (FaSScinate), and the data suggested this drug played a role in the IL-6 pathway in SSc-ILD and treatment of early SSc with elevated C-Reactive protein (CRP) and that it led to the stabilization of the FVC% in the tocilizumab group vs. a clinically meaningful decline in the placebo group over 48 weeks [48]. In this view, the phase 3 double-blind randomized placebo-controlled study (FocuSSced) of TCZ enrolled 210 early dcSSc patients. Similarly, a reduced FVC decline was seen in the TCZ group (difference between groups 4.2 (95% CI 2.0–6.4) favoring TCZ; *p* = 0.0002), with a trend for a lower rate of patients requiring rescue immunosuppressive therapy for ILD indication (*p* = 0.08) [49].

Although the primary (skin) endpoint was not met, both trials showed some efficacy and a good safety profile for using TCZ in SSc and evidenced a potential benefit of treating subclinical ILD patients with high risk features (early dcSSc, and elevated CRP) [50].

### 4.6. Abatacept

Abatacept is a recombinant fusion protein that inhibits T cell activation. An observational study was carried out on 20 patients with SSc-associated polyarthritis and myopathy to evaluate the safety and efficacy of TCZ. However, despite having a good safety profile, there was no change in lung fibrosis in patients treated with abatacept [51]. A similar, more recent, observational experiment from the EUSTAR group, which included 27 SSc patients (15 with ILD), confirmed the good safety profile of the drug as well as a beneficial effect on joint and muscle disease. In addition, a possible positive effect was also seen for skin fibrosis (despite the lack of a control group), while no significant change in lung function was detected [56]. Finally, a recent phase II multicentre double-blind placebo-controlled trial of abatacept in early dcSSc showed a trend for a significant lower decline of FVC% predicted (mean difference 2.79, 95% CI −0,69–6,27, *p* = 0,11 favouring Abatacept), although the change in skin fibrosis as a primary endpoint was, again, not met [57]. Despite this, there is a promising potential for Abatacept in SSc, which could be investigated in a phase III study.

## 5. Other Treatment Options 

### 5.1. Immunoglobulins

A randomized control trial assessing the change in skin fibrosis as primary endpoint failed to demonstrate a significant beneficial reduction of modified Rodnan skin score in Intravenous immunoglobulin (IVIg) administration versus placebo [71]. Different authors have shown the beneficial effect of the use of IVIg in SSc patients with arthritis [72] and inflammatory myopathy [73], and they have demonstrated some potential benefit on early-stage ILD, with the authors reporting a regression in ground glass opacity, septal thickening, and a full recovery of lung function [74]. A randomized phase II trial (NCT04137224) is currently testing a possible similar effect and the safety of subcutaneous immunoglobulins and IVIg [75].

### 5.2. Autologous Hematopoietic Stem Cell Transplant

Hematopoietic stem cell transplant (HSCT) is an emerging treatment option, aimed at regenerating the patient’s immune system [43,76]. It is based on the use of high-intensity immunosuppression (conditioning regimen) aimed at a strong reduction/eradication of the “auto-reactive” immune system, followed by a re-population with antigen-naïve T cells previously isolated from the same individual [76]. It has been proposed for patients with dcSSc (with or without SSc-ILD) that is severe and refractory to standard therapy, who will probably benefit from the procedure but are more unlikely to develop post-transplant complications [44,77]. Indeed, improved survival compared to CYC has been reported by three trials, i.e., Autologous Stem Cell Systemic Sclerosis Immune Suppression Trial (ASSIST), Autologous Stem Cell Transplantation International Scleroderma trial for (ASTIS), and Scleroderma: Cyclophosphamide Or Transplantation trial (SCOT). Moreover, there was an improvement in skin thickening and FVC, as well as quality of life [38,39,45]. In addition to these results, an observational analysis of ILD extent in SSc patients receiving HSCT versus CYC was recently published, showing significant reduction in total ILD extent and, in particular, in the extent of ground glass opacifications, which were not seen in the CYC group [46]. With this promising background, a phase III randomized clinical trial (NCT044644) is currently testing upfront HSCT versus intravenous CYC induction followed by maintenance with MMF in early dcSSc, including pulmonary endpoints [78].

### 5.3. Lung Transplant

Lung Transplant is a life-saving option and remains a therapy for appropriately selected candidates with treatment-refractory lung disease [47,79]. An early referral should be made for advancing disease so as to provide these patients with a multi-disciplinary evaluation before transplant is considered an option. A few recent studies have demonstrated an increase in survival after lung transplantation [47,79]. 

### 5.4. Non-Pharmacologic Therapy

SSc-ILD should be managed by a multidisciplinary team [80]. Among non-pharmacologic options, pulmonary rehabilitation is aimed at improving lung function [81]; in particular, when an SSc-ILD patient is being considered for a transplant, pulmonary rehabilitation is a necessary step in their evaluation [81]. Furthermore, supplemental oxygen should be given whenever deemed necessary. 

## 6. Anti-Fibrotic Therapies

Nintedanib is an intracellular tyrosine kinase inhibitor approved for the first time for the treatment of idiopathic pulmonary fibrosis (IPF) [82,83]. Its pharmacological effect covers numerous pathophysiological pathways, such as fibroblast activation, myofibroblast accumulation, and fibrogenic cytokine and growth factor expression. The increasing number of national and international authorities giving approval for treatment with the antifibrotic agent nintedanib to slow down the rate of decline of pulmonary function in SSc-ILD patients is opening up a new era [84]. The results of the recently published Safety and Efficacy of Nintedanib in Systemic Sclerosis (SENSCIS) trial supported the decision [20]. The SENSCIS trial, a double blind, randomized, placebo-controlled trial, evaluated the efficacy and safety of oral nintenadib (150 mg bid) treatment in patients with SSc-ILD for at least 52 weeks [20]. It reported that almost 50% of the subjects had dcSSc; a similar percentage was on a stable dose of MMF and HRCT evidenced fibrosis in at least 10% of the lungs (the latter as per study inclusion criteria). In this trial, patients with SSc-ILD treated with nintedanib showed a significantly lower rate of annual FVC decline than those receiving placebo, despite no significant improvement or benefit in any of the other organ manifestations. Although the change in FVC was small (absolute mean decline mean −52.4 mL per year in the nintedanib versus −93.3 mL per year in the placebo group), the mean decline reached a previously shown value of minimal clinically important difference (MCID) in the placebo group, but not in the nintedanib treated population [85]. This beneficial effect on FVC preservation was seen both in MMF and non-MMF co-treated patients, with a numerically lower decline in patients receiving the combination treatment [58]. In the SENSCIS study, Nintedanib showed a safety profile similar to the side effects seen in IPF, particularly affecting the gastrointestinal tract (75.7% of treated patients manifested diarrhea) and requiring dose-adjustment/temporary interruption in almost half of the treated patients [59]. Interestingly, the safety profile was similar in patients receiving or not receiving co-treatment with MMF, which itself carries a gastro-intestinal burden in terms of adverse events [58]. 

With a similar multi-target pathogenetic activity, pirfenidone (a pyridone showing both anti-inflammatory and anti-fibrotic effects) is another antifibrotic agents approved for the management of IPF patients [86]. The initial compassionate use in selected patients with SSc–ILD showed that the drug was well tolerated and, although it did not improve survival, it did stabilize the effects of progressive pulmonary fibrosis [60,61]. Recently, the Safety and Tolerability of Pirfenidone in Participants with Systemic Sclerosis-related Interstitial Lung Disease (LOTUSS) study, a phase II, open-label, randomized, 16-week study, assessed the safety and tolerability of pirfenidone in SSc–ILD patients. The drug was reported to have an acceptable tolerability profile that was not affected by concomitant treatment with MMF, although data as to its efficacy is not yet available [62]. Indeed, pirfenidone can be associated with adverse events of the gastrointestinal system and the skin in patients with IPF, two organs very frequently involved in SSc-ILD. Sometimes these adverse events can lead to drug discontinuation [63]. Given the promising effect in the stabilization of SSc-ILD, the drug is now tested versus placebo in SSc-ILD patients receiving MMF as a background immunosuppressive therapy in a placebo-controlled multi-center double blind randomized SLS study III [87]. 

## 7. Conclusions

Although ILD is a common finding in SSc, currently there is a paucity of detailed data to help in predicting which subsets of patients will or will not develop organ and potentially life-threatening disease. Despite this, the potential risk of morbidity and mortality supports the need for a thorough and early monitoring of the signs and symptoms of the development and progression of ILD. 

At time of writing, the standard of care includes the use of CYC and MMF (which have only provided modest improvements in FVC) and Nintedanib (which is not available worldwide). Preliminary data on newer therapies, like biologics, stem cell transplant, and other anti-fibrotics suggest improved efficacy and safety profiles compared to those obtained with conventional immunosuppressive therapy. 

Following the SLS II trial, a Delphi consensus treatment algorithm advocated MMF as first-line treatment of SSc-ILD and suggested that second-line treatment should include CYC or rituximab as an induction therapy, followed by MMF as a maintenance therapy [88]. 

A more recently published European consensus on SSc-ILD identification and management stressed the importance of different factors guiding treatment initiation, including speed of disease progression, survival rate, response rate after previous treatment, prolongation of time to progression, speed of improvement of patients’ symptoms, safety and tolerability, scientific evidence of efficacy, and impact on quality of life [15]. 

In addition, disease severity and speed of progression could be the main drivers of treatment escalation [15]. Although consensus and recommendations are nowadays available, these do not fully cover the different clinical scenarios, in particular regarding time to initiation and a possibly more effective treatment protocol. In this context, SSc experts still relay on their clinical experience and take into account the different abovementioned factors to guide their decision in a patient-tailored, customized treatment regimen [89], possibly informed also by molecular biomarkers [90].

Although a substantial amount of evidences in SSc-ILD management resemble IPF, SSc-ILD is not IPF, as patients with SSc-ILD have a systemic disease. It has been hypothesized that future therapeutic options may be provided by targeting the self-perpetuating fibrosis [91], although whether an early (immunosuppressive) aggressive treatment will lead to a modification of disease progression and prevention of irreversible lung damage remains a question of debate. In this context, limiting fibrogenesis by the use of antifibrotic therapy and controlling inflammation/immunological abnormalities through immunosuppressants could well become the new paradigm of treatment in SSc-ILD. If available and well-tolerated, a combination regimen with immunosuppression and anti-fibrotic may allow a multi-target treatment and, potentially, a multiple organ/system benefit. Specifically, immunosuppression could also be personalized according to non-ILD organ complications such as cutaneous involvement, arthritis, myositis, and cardiomyopathy.

Clearly, there is a need for guidance in the new treatment regimens, in particular regarding the use of upfront or add-on combination treatment with immunosuppressants/antifibrotic, which could be the possible second/third level option in case of treatment failure. Hopefully, our understanding of the pathogenesis of SSc-ILD will evolve, along with the development of specific therapies for the organ systems affected by this disease, thus improving patients’ survival, function, and quality of life [92,93].

## Figures and Tables

**Table 1 pharmaceuticals-14-00154-t001:** Summary of current treatment options for systemic sclerosis related interstitial lung disease.

Drug	Study Designs	Pulmonary Parameters Tested
Cyclophosphamide [16,17,31,32,33,34,35,36,37,38,39]	RCT, OS	FVC, DLCO, ILD progression, HRCT disease extent, PROs
Mycophenolate Mofetil [17,33,40,41,42]	RCT, OS	FVC, DLCO, ILD progression, HRCT disease extent, PROs
Azathioprine [31,34,35,36]	RCT, CS/CR	FVC, DLCO, PROs
Autologous Haematopoietic Stem Cell Transplantation [38,39,43,44,45,46,47]	RCT, CS/CR	FVC, DLCO, Total Lung capacity, Vital Capacity, HRCT disease extent
Tocilizumab [48,49,50,51]	RCT, CS/CR	FVC, DLCO, HRCT disease extent, PROs
Rituximab [37,52,53,54,55]	RCT, OS, CS/CR	FVC, DLCO, PROs
Abatacept [51,56,57]	RCT, CS/CR	FVC, DLCO, PROs
Nintedanib [20,58,59]	RCT	FVC, DLCO, PROs
Pirfenidone [60,61,62,63]	RCT, CS/CR	FVC, DLCO, PROs

DLCO = diffusion lung capacity of carbone monoxyde; FVC = forced vital capacity; HRCT = high resolution computed tomography; ILD = interstitial lung disease; OS = observational study; PROs = patient reported outcomes; RCT = randomized clinical trial; CR/CS = case report/case series.

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
