# Peer review of "The Treatment of Lung Involvement in Systemic Sclerosis"

_pharmaceuticals, 2021, doi:10.3390/ph14020154_

Round 1
Reviewer 1 Report
Can the authors comment on the fact that although Nintedanib did reduce the rate of decline of pulmonary function in the SENSCIS trial, the absolute change n FVC % was small.
This would benefit from a table summarising the current therapies in use or possible use in SSc ILD.
HSCT section gives a great overview of the clinical trial of HSCT and clearly demonstrates benefit- at least in a subset of patients. Could the authors add some details on possible mechanism (s) that underpin the therapeutic effect.
Reference 30-32 are incomplete and must include T cells in systemic sclerosis: a reappraisal Rheumatology 2012 51: 1540 O'Reilly et al.
Final reference 85 should be supplemented with Current and Potential new targets in systemic sclerosis: A new hope Current Rheumatology reports 2020 22 Hinchcliff et al.
Line 149 typo ''Goof'' mean ''Good''
Author Response
We thank the reviewers for his comments, to whom we reply below.
REVIEWER 1
Can the authors comment on the fact that although Nintedanib did reduce the rate of decline of pulmonary function in the SENSCIS trial, the absolute change n FVC % was small.
Thanks for your comment, we implemented the paragraph accordingly.
Page 9, line 25-28: “. Although the change in FVC was small (absolute mean decline mean -52.4 ml per year in the nintedanib versus -93.3 ml per year in the placebo group), the mean decline reached previously shown value of minimal clinically important difference (MCID) in the placebo group, but not in the nintedanib treated population [79],”
This would benefit from a table summarising the current therapies in use or possible use in SSc ILD.
Thanks for your suggestion, a summary table was added.
HSCT section gives a great overview of the clinical trial of HSCT and clearly demonstrates benefit- at least in a subset of patients. Could the authors add some details on possible mechanism (s) that underpin the therapeutic effect.
Thanks for your suggestion, we further implemented the section on HRST with more details.
Page 8, line 14-18: “Hematopoietic stem cell transplant (HSCT) is an emerging treatment option, aiming at regenerating the patient’s immune system [63,64]. It is based on the use of high-intensity immunosuppression (conditioning regimen) aiming to a strong reduction/erase of the “auto-reactive” immune system, followed by a re-population with antigen-naïve T cells previously isolated from the same individual [63].”
Reference 30-32 are incomplete and must include T cells in systemic sclerosis: a reappraisal Rheumatology 2012 51: 1540 O'Reilly et al.
Final reference 85 should be supplemented with Current and Potential new targets in systemic sclerosis: A new hope Current Rheumatology reports 2020 22 Hinchcliff et al.
Thank you for suggesting this very important refences, to further support our statements.
Line 149 typo ''Goof'' mean ''Good''
The typo was corrected, thank you!
Reviewer 2 Report
The manuscript represents a state or the art review on the treatment of lung involvement in systemic sclerosis. The manuscript is well organized, easy to follow, concise. As stated before, I do not feel qualified to judge about the English language and style; however, a few typos should be corrected (e.g. page 3; lines 141-141). Many abbreviations are not properly defined (e.g. cyclophosphamide).
Scientific concerns:
Page 2, line 71: The Authors indicate a decline in FVC levels > 10% as a defining feature of progressive disease but fail to indicate the time frame of such decrease.
Page 6, line 256: The sentence "The drug [pirfenidone] was adminitered as a compassionate treatment in eight IPF patients and two patients with SSc-ILD" must be wrong, since prifenidone is approved for IPF; furthermore, ref 75 is the ASCEND randomized controlled trial by King at al.
Author Response
We thanks the reviewer for the valuable comments, to whom we reply below.
REVIEWER 2
The manuscript represents a state or the art review on the treatment of lung involvement in systemic sclerosis. The manuscript is well organized, easy to follow, concise. As stated before, I do not feel qualified to judge about the English language and style.
however, a few typos should be corrected (e.g. page 3; lines 141-141).
All identified typos were corrected, thank you.
Many abbreviations are not properly defined (e.g. cyclophosphamide).
Abbreviations were checked and corrected, repeating the full work only as paragraph titles.
Scientific concerns:
Page 2, line 71: The Authors indicate a decline in FVC levels > 10% as a defining feature of progressive disease but fail to indicate the time frame of such decrease.
Thanks for pointing this out. We clarified it in the text and added appropriate references to support.
Page 4, line 10-15 “It is a must to detect the subset of clinical ILD patients with progressive disease, defined as a decline in Forced Vital Capacity (FVC) levels of >10% from baseline or a ≥ 5% to < 10% relative decline in FVC and a ≥ 15% relative decline in Diffusion Lung Capacity of Carbon monOxyde (DLCO) over 12 months [28]. Despite this cut-off being proposed for clinical trials and applied also in clinical practice, although smaller changes may also be of clinical importance, in particular worsening symptoms attributable to ILD [25,26,29].”
Page 6, line 256: The sentence "The drug [pirfenidone] was adminitered as a compassionate treatment in eight IPF patients and two patients with SSc-ILD" must be wrong, since prifenidone is approved for IPF; furthermore, ref 75 is the ASCEND randomized controlled trial by King at al.
The paragraph about Pirfenidone was edited, following your indications, Thank you.
Page 7, line 7-14: “With a similar multi-target pathogenetic activity, pirfenidone (a pyridone showing both anti-inflammatory and anti-fibrotic effects) is another antifibrotic agents approved for the management of IPF patients [82]. the initial compassionate use in selected patients with SSc–ILD, showed that the drug was well tolerated and, although it did not improve survival, it did stabilize the effects of progressive pulmonary fibrosis [82]. Further studies reported that pirfenidone has a stabilizing effect on ILD in SSc patients although its efficacy in SSc still remains to be clarified [83,84].”